# New Evidence on BPA’s Role in Adipose Tissue Development of Proinflammatory Processes and Its Relationship with Obesity

**DOI:** 10.3390/ijms24098231

**Published:** 2023-05-04

**Authors:** Jorge Enrique González-Casanova, Valmore Bermúdez, Nelson Javier Caro Fuentes, Lissé Chiquinquirá Angarita, Nelson Hernando Caicedo, Jocelyn Rivas Muñoz, Diana Marcela Rojas-Gómez

**Affiliations:** 1Facultad de Ciencias de la Salud, Instituto de Ciencias Biomédicas, Universidad Autónoma de Chile, Santiago 8910060, Chile; jorge.gonzalez@uautonoma.cl; 2Centro de Investigaciones en Ciencias de la Vida, Universidad Simón Bolívar, Barranquilla 080002, Colombia; 3Centro de Investigación Austral Biotech, Facultad de Ciencias, Universidad Santo Tomás, Avda. Ejército 146, Santiago 8320000, Chile; 4Escuela de Nutrición y Dietética, Facultad de Medicina, Universidad Andres Bello, Sede Concepción, Talcahuano 4260000, Chile; lisse.angarita@unab.cl; 5Departamento de Ciencias Biológicas, Bioprocesos y Biotecnología, Facultad de Ingeniería, Diseño y Ciencias Aplicadas, Universidad Icesi, Calle 18 No. 122–135 Pance, Cali 760031, Colombia; 6Escuela de Nutrición y Dietética, Facultad de Medicina, Universidad Andres Bello, Santiago 8370321, Chile

**Keywords:** bisphenol A, inflammation, adipose tissue, endocrine disruptor

## Abstract

Bisphenol A (BPA) is a xenobiotic with endocrine disruptor properties which interacts with various receptors, eliciting a cellular response. In the plastic industry, BPA is widely used in the production of polycarbonate and epoxy-phenolic resins to provide elastic properties. It can be found in the lining of canned foods, certain plastic containers, thermal printing papers, composite dental fillings, and medical devices, among other things. Therefore, it is a compound that, directly or indirectly, is in daily contact with the human organism. BPA is postulated to be a factor responsible for the global epidemic of obesity and non-communicable chronic diseases, belonging to the obesogenic and diabetogenic group of compounds. Hence, this endocrine disruptor may be responsible for the development of metabolic disorders, promoting in fat cells an increase in proinflammatory pathways and upregulating the expression and release of certain cytokines, such as IL6, IL1β, and TNFα. These, in turn, at a systemic and local level, are associated with a chronic low-grade inflammatory state, which allows the perpetuation of the typical physiological complications of obesity.

## 1. Introduction

Bisphenol A (BPA) is a chemical widely used as a component for polycarbonate and epoxy-phenolic resin production which is highly resistant to heat, providing elasticity to plastic materials. It can be found in the lining of canned foods, certain plastic containers, thermal printing papers, composite dental fillings, and medical devices, among other things. Therefore, it is a compound frequently found, on a daily basis, in contact with the human organism [1].

Evidence suggests that BPA could be a contributing factor in the global obesity epidemic and in non-communicable chronic diseases, making part of the so-called obesogenic and diabetogenic compounds [2,3].

BPA has been incorporated in the food chain, generating increased interest in its safe use by the public, as well as the scientific community. Therefore, various exploratory scientific studies on its consequences at the cellular and physiological levels in different body systems and tissues have arisen. Some of these studies regarded safe intake levels, yet experimental and epidemiological results supported the fact that low doses of this compound have the capacity to affect, with negative repercussions, the endocrine system at the metabolic level. In absolute terms, adult human exposure to BPA is low; however, chronic exposure can take place depending on BPA-carrying vectors [4].

Based on the latest scientific findings, the European Food Safety Authority (EFSA) proposed the reassessment of the risks of BPA in food and considered significantly lowering the tolerable daily intake (TDI) for BPA from that given in its previous assessment in 2015, and, in 2019, a new TDI of 0.04 ng/kg bw/day was established to replace the previously proposed TDI of 4 µg/kg bw/day. This decrease in the TDI was based on evaluation studies published between 2013 and 2018, mainly due to the results that showed the adverse effects of BPA on the immune system [5].

BPA routes of exposure to the human organism include oral intake, inhalation, contact with skin and eyes, and by fetal–maternal transmission, since BPA can cross the placental barrier [6] (Figure 1). 

It is estimated that 90% of exposure to BPA is through food, and, usually, only 5% corresponds to inhalation or through dental materials or dermal exposure [7]. However, exposure to BPA through the skin produces a longer half-life of this compound in the body when compared to the half-life of BPA taken in through oral exposure [8,9].

When BPA enters the body, it is metabolized in the liver, and a BPA–glucuronate conjugate is formed, a metabolite that has been shown to be stable and valid as a biomarker [10]. Due to its lipophilic nature, BPA has the ability to be deposited in human and animal adipose tissue [11]. BPA is excreted in the urine in a relatively short time, and virtually all the BPA that enters the body is eliminated within 48 h without a significant amount of retention in body tissues [4].

Regarding BPA levels detected in human biological samples, levels ranging from picomolar to nanomolar ranges have been reported [12]. Table 1 and Table 2 illustrate the most recent levels reported in human plasma and urine in recent years. 

## 2. BPA as an Endocrine Disruptor

BPA is a xenobiotic with endocrine disruptor characteristics which interacts with various nuclear receptors, producing a specific cellular response. It has been shown that BPA has biological activity, binding to various receptors such as estrogen α and β receptors (ERs), G-protein-associated receptor GRP30, androgen receptors (ARs), thyroid hormone receptors (TRα and β), estrogen-related receptor gamma (ERRγ), and glucocorticoid receptor (GR) (Figure 2).

### 2.1. Estrogen α and β Receptors

In 1936, Dodds and Lawson demonstrated BPA’s estrogenic properties in vivo [44]. It was shown that this compound acts as a 17-beta estradiol (E2) steroidal hormone; however, its affinity is 1000 to 10,000 times lower in comparison with the E2 physiological receptor [45]. Classical estrogen receptors are most commonly identified as nuclear receptors, although these same proteins appear to have membrane-bound variants. BPA exerts effects on either nuclear or membrane-localized ERs of the α type. Additionally, BPA has the capacity to increase progesterone receptor expression, yet with a capacity 2000 times lower than estradiol. The physiological consequences of BPA-induced modulations are diverse and depend on the tissue examined and the study model [45].

### 2.2. GPR30 Receptor

The GPR30 is a more recently described class of estrogen receptor belonging to the G-protein-coupled receptors family, which mediates E2-elicited cellular signals. These receptors regulate cellular responses through second messengers and elicit a faster response in comparison with intracellular steroid hormone receptors. For GPR30, the second messenger is cyclic AMP and elicits cellular signals involved in extracellular-signal-regulated kinase 1/2 (ERK1/2) [46]. 

### 2.3. Androgen Receptor (AR)

ARs bind steroid hormones through nuclear receptors. This type of receptor is expressed in all male and female organs, and, as with ERs, it shares similar mechanisms of action, as well as its location within the cell. Once an AR interacts with its ligand, it travels to the nucleus and forms homodimers. Within the nucleus, it interacts with *androgen response element* promoter regions, modifying the expression of androgen-responsive genes.

In silico studies have reported the ability of BPA to bind to multiple sites on the AR surface through hydrophobic interactions. In contrast, in vivo studies have shown its antagonistic activity to AR [47], where BPA acts as a competitive inhibitor [48]. Moreover, it has the ability to inhibit testosterone-promoted translocation of the receptor to the nucleus [49]. The anti-androgenic effects of BPA on male reproductive function may be mediated by different mechanisms involving stabilization, ligand-induced AR heat shock protein 90 dissociation, and nuclear translocation of the AR receptor [50].

## 3. Inflammatory Processes in Adipose Tissue

Adipose tissue adapts to the different nutrient intake changes in the body to maintain metabolic and energy homeostasis. Nutrient deficiency or excess alters the size and physiology of fat cells and causes adaptive cellular events known as “tissue remodeling” [51]. 

Remodeling of adipose tissue taking place in obesity sometimes involves chronic proinflammatory reactions with systemic repercussions known as lipo-inflammation [52]. This remodeling is closely associated with metabolic disorders, such as insulin resistance, type 2 diabetes mellitus, cardiovascular diseases, and resistance to catecholamines. Furthermore, inflammation level is proportionally related to metabolic disease severity [53]. Inflammation can be explained as an adaptive response to excess energy or adverse cellular conditions or tissue stress, stimulating angiogenesis to prevent hypoxia. Additionally, inflammation increases insulin resistance, a cellular mechanism that limits the rate of energy accumulation at the muscle or adipose tissue level. 

The inflammation process represents an ordered sequence of cellular events that maintains tissue homeostasis; hence, it is a protective response whose function involves the destruction or dilution of the damaging agent or injured tissues or the removal of dead cells [54]. 

Inflammation can occur acutely with edema formation and leukocyte migration, or inflammation may be chronic with the presence of M1-type proinflammatory macrophages and lymphocytes, along with proliferation of blood vessels and connective tissue that occurs over a prolonged period [55]. 

The typical complications of obesity, and, by extension, of adipose tissue, are closely related to chronic inflammation, characterized by proinflammatory molecule secretion by macrophages and adipocytes, such as secretion of interleukin 6 (IL6), tumor necrosis factor-α (TNF-α), monocyte chemoattractant protein-1 (MCP-1), and interleukin 1β (IL1β), and activation of nuclear factor κβ (NF-κβ), the master regulator of inflammation [53]. 

## 4. Influence of BPA on Inflammatory Signals in Adipose Tissue 

Scientific evidence has shown a potential role for BPA in the development of metabolic disorders associated with inflammation; however, the mechanisms involved are still unclear.

BPA promotes the expression and release of certain proinflammatory cytokines, resulting in a constant low-grade inflammatory state at local and systemic levels [56]. Additionally, BPA has been reported to interfere with adipogenic processes [57]; therefore, it is important to establish the existence of a possible association between inflammatory events during adipocyte differentiation and BPA’s influence. Several investigations have shed light on BPA’s influence on inflammatory processes during adipogenesis. For example, experimental results in 3T3-L1 cell lines exposed to 1 nM BPA showed an elevation of leptin, IL6, and IFNγ during the final stage of the differentiation process towards adipocytes [58]. Similar effects were obtained by Valentino et al. [59] in cultures of adipocytes derived from human subcutaneous adipose tissue and in cultures of adipocytes from the 3T3-L1 cell line exposed to 1 nM BPA. In these experiments, higher values of IL6 and IFNγ were observed when compared to those of the control group (0.98 ± 0.03 vs. 5.27 ± 0.7 pg/mL and 0.25 ± 0.09 vs. 0.86 ± 0.05 pg/mL, respectively).

More recently, Longo et al. [60] demonstrated that exposure for a period of 8 days during the differentiation process of 3T3-L1 cells increased the mRNA levels of IL6, INFγ, TNFα, MCP1, and IL1β. Interestingly, the proinflammatory effect observed after BPA exposure was reversible, as removal of BPA from the culture media resulted in decreased expression levels of proinflammatory cytokines similar to the levels found in the control group. 

Furthermore, experiments were carried out in a murine model with 5-week-old male and female C57BL/6J mice exposed to four doses of BPA (5, 50, 500, and 5000 µg/kg/day) by oral intake for 30 days and fed chow diets (DC) or high-fat diets (DAG) [61]. Under these conditions, it was shown that BPA, starting at daily 5 µg/kg/day concentrations, in a non-monotonic dose-response fashion, together with a DC diet, caused an increase in body weight and fat mass. This increase was not observed in DAG-fed mice. Additionally, in white adipose tissue, increased expression of F4/80, Cd11c, and Mcp1 was observed in male mice treated with BPA and fed with DC, suggesting macrophage migration to adipose tissue. Increased *IL6*, *TNFα*, *IL1β*, *IFNγ*, and *iNos2* mRNA expression was also observed in female and male mice exposed to the highest concentrations of BPA (500 and 5000 µg/kg/day) and fed DC. DAG-fed mice did not exhibit these effects after BPA exposure. These results, therefore, suggest that BPA’s effect may depend on diet composition and caloric intake, altering the susceptibility to obesity or promoting a proinflammatory profile in adipose tissue depending on the proportion of fatty dietary components. 

Additionally, this same group demonstrated that, in normal weight women (body mass index < 23.0 kg/m^2^), BPA concentration was associated with an increase in circulating inflammatory factors, including leptin and TNFα; however, in lean male subjects and in both sexes of overweight/obese individuals (body mass index > 25.0 kg/m^2^), no such correlation was identified [61]. 

In a population in India, BPA and its relationship with nutritional status in patients with type 2 diabetes mellitus (n: 150) versus a healthy individual control group with normal glucose tolerance (n: 150) were studied. In the Indian study, it was found that plasma BPA levels were directly related to body mass index, waist circumference, and leptin levels. Nevertheless, BPA showed an inverse relationship with plasma adiponectin levels [62]. In the control group, these correlations were not observed. When analyzing the correlation between proinflammatory molecule levels and BPA’s plasma concentration, no significant association was observed between IL6, TNFα, and IL1β plasma levels and BPA. These results are partly explained by the ability of BPA to accumulate in adipose tissue due to its lipophilic nature. Therefore, plasma levels do not reflect the physiological action of this endocrine disruptor at the tissue level. These findings are in agreement with what was reported by the National Health and Nutrition Examination Survey (NHANES) results [63], where lower-than-expected urinary values were observed, suggesting that BPA can be stored in certain body compartments, such as adipose tissue, from which it is slowly released [64]. Reports of the presence of BPA in adipose tissue have already been described in ranges between minimum values of 4.65 ng/g of tissue and maximum values of 50 ng/g of tissue (Table 3). 

Unlike the previously mentioned studies, where BPA was shown to have proinflammatory effects on adipose tissue cells, other studies in exposed human subcutaneous adipose tissue reported that BPA can reduce proinflammatory cytokine gene expression, such as expression of *IL6*, *IL1β*, and *TNFα* at BPA’s supra-physiological concentrations of 1–104 nM for 24 h or 72 h [65]. These results are interesting, since, contrary to what was expected, there was a reduction in proinflammatory cytokines in adipose tissue, in contrast to the hypothesis that BPA is an inflammatory agent. The authors raised the possibility that, although, in adipocytes, BPA can stimulate the expression of inflammatory markers, it has also been shown that this endocrine disruptor can, in turn, decrease the expression of proinflammatory cytokines in macrophages [66,67]. Therefore, it is feasible that, under the experimental conditions of Ahmed’s group study [65], samples collected from complete adipose tissue displayed a different behavior in comparison with BPA’s action in isolated adipocytes and under the concentrations of BPA used. The complexity of the interaction between BPA and the diverse cellular component of adipose tissue reveals the difficulty of extrapolating results from cultures of isolated adipocytes, or from adipose tissue, and even from experiments in animal models.

**Table 3 ijms-24-08231-t003:** BPA levels found in samples of human adipose tissue.

References (Year)	Type of Participants (n: Number of Participants)	Levels Found (ng/g Tissue)	Detection Method	Country
Venisse et al. (2019) [68]	Patients during breast or prostate surgery (n: 5).	Range: 1.19–8.73.	LC-MS/MS	France
Artacho-Cordón et al. (2017) [64]	Patients undergoing trauma surgery (n: 14).	Mean: 0.60.	HPLC	Spain
Reeves et al. (2018) [69]	- Breast cancer mastectomy patients: (n: 36); - Control group: reduction mammoplasty patients (n: 14).	Mean (SD): - Breast cancer mastectomy patients: - Among all samples 0.19 (0.35); - Among samples with detectable BPA 0.71 (0.31); - Control: - Among all samples 0.26 (0.37); - Among samples with detectable BPA 0.66 (0.27).	HPLC-ESI-MS/MS	USA
Keshavarz-Maleki et al. (2021) [21]	- Breast cancer mastectomy patients (n: 41); - Control group: mammoplasty patients (n: 11).	- Cancerous patients 4.20 ± 2.40; - Control group: 1.80 ± 1.05.	ELISA assay	Iran
Salamanca-Fernández et al. (2020) [70]	Sub-cohort of the Spanish European Prospective Investigation into Cancer and Nutrition (EPIC) (n: 4812): - Breast cancer cases (n: 547); - Prostate cancer cases (n: 575); - Sub-cohort participants (n: 3690).	Geometric mean: - Breast cancer sub-cohort: 1.10; - Breast cancer cases: 1.12; - Prostate cancer sub-cohort: 1.29; - Prostate cancer cases: 1.33.	UHPLC-MS/MS	Spain

HPLC: high-pressure liquid chromatography; HPLC-ESI-MS/MS: high-performance liquid chromatography/electrospray ionization tandem mass spectrometry; LC-MS/MS: high-performance liquid chromatography coupled with tandem mass spectrometry; UHPLC-MS/MS: ultra-high-performance liquid chromatography with tandem mass spectrometry; ELISA: enzyme-linked immunosorbent assay.

## 5. BPA’s Inflammatory Action and Cellular Mechanisms Involved 

Cimmino and collaborators [71] studied in adipose tissue possible BPA mechanisms associated with alteration of inflammatory mediator expression. Specifically, they evaluated GPR30’s role in the BPA-mediated inflammatory response. GPR30 is involved in estrogen-dependent rapid signaling as well as transcriptional activation, which are not dependent on classical nuclear estrogen receptors. Interestingly, BPA has the ability to bind to GPR30 and, consequently, activate cell signaling. The experiments were carried out on mature adipocytes or stromal vascular fraction cells cultured with 0.1 nM BPA (low dose) obtained from the breast tissue of overweight women. Under these experimental conditions, an elevation in IL8, MCP1α, IL6, TNFα, and IL1β was observed. On the contrary, expression of the anti-inflammatory interleukin IL10 was decreased. The GPR30 mechanism of action was verified with specific receptor agonists and antagonists since similar results were obtained when 100 nM of G1 agonist was used. In addition, this effect was reversed with 1 µM G15, a selective GPR30 antagonist. Likewise, under BPA’s induction, an increase in the expression of GPR30 was observed. The same study [71] also demonstrated that IL8 is a possible mediator of BPA’s effect on adipose tissue, favoring cell proliferation from stromal vascular fraction and adipocyte expansion, because IL8 inhibition reversed the effect of BPA on adipose tissue cell growth. 

IL8 is a proinflammatory cytokine related to chronic inflammatory processes in adipose tissue from obese patients [72,73,74]. Therefore, these studies suggest that IL8 plays a role as a secondary effector of BPA.

Reactive oxygen species (ROS) are key signaling molecules in the progression of inflammatory disorders [75], and BPA is postulated as an enhancer of ROS generation [76]. In this context, the Artacho-Cordón group [64] demonstrated, in adipose tissue samples obtained from 144 patients from southern Spain, a significant inverse association between BPA values and glutathione reductase activity in addition to increased oxidized glutathione. These results suggest for BPA at adipose tissue local level a potential role in redox balance alteration.

Recently, in a human case-control study, Hong et al. [32] demonstrated a positive correlation between urinary BPA levels and obesity and fasting insulin and glycemia, as well as body mass index (BMI). Interestingly, BPA was also positively associated with elevated plasma IL-17A, a cytokine that has been shown to be involved in chronic inflammation [77]. Similarly, an increased accumulation of IL-17A was observed in the adipose tissue of patients undergoing bariatric surgery, where IL-17A levels were higher in patients with greater BMI upon BPA exposure. These results were corroborated in a high-fat-diet (HFD)-induced obese mouse model where mice were exposed to BPA, resulting in adipose tissue with an increase in proinflammatory M1 macrophages TNF-α and IL-1β. Moreover, a significantly higher proportion of Th17 cells, as well as an increase in adipose tissue IL-17A, was observed. In agreement with these results, HFD IL-17A^−/−^ mice were capable of reversing the inflammation in adipose tissue even when exposed to BPA [32]. Collectively, these results evidence IL-17A’s potential role by establishing an association between BPA exposure, the risk of becoming obese, and chronic inflammation.

## 6. Experiments in Fetal Programming: Effect of BPA on Adipose Tissue and Inflammation during Gestation

More than 90% of pregnant women have detectable urinary BPA concentrations, indicating widespread exposure, including during the prenatal period [78]. There is a consensus that during the perinatal period, when the development of the organism takes place, a critical window for BPA exposure ensues where the immune system and metabolic functions, including body weight regulation, can be affected [79]. 

Studies with BPA in the offspring of animal models during gestation have shown its effects on adipose tissue. A prenatal response study in sheep to BPA doses of 0.05, 0.5, or 5 mg/kg/day demonstrated that adult offspring (21-month-old females) presented elevated CD68 expression in subcutaneous adipose tissue but not in visceral adipose tissue [80]. Where CD68 has been described as a marker of macrophage infiltration, CD68 elevation was noteworthy, and, thus, possible macrophage infiltration was limited to subcutaneous adipose tissue, suggesting BPA’s selectivity to fat deposits, a characteristic that has been reported for other organic contaminants. 

However, other studies showed different results regarding the behavior of subcutaneous adipose tissue in the presence of BPA. Another study, evaluating sheep exposed to BPA with daily doses of 0.5 mg/kg during their fetal growth, demonstrated development of visceral adipose tissue hypertrophy at 21 months after birth [81]. This treatment allowed free BPA concentrations of 2.62 ± 0.52 ng/mL collected from the umbilical artery on day 90 of fetal life with values within the range observed in human umbilical cord blood concentrations at mid-gestation [27]. In addition, differential gene expression was evaluated by tissue type (subcutaneous vs. visceral) and by treatment (control vs. prenatal BPA group). The results showed, when tissue type was compared, a positive regulation of inflammation, oxidative stress, and adipose differentiation genes in subcutaneous adipose tissue compared to in visceral adipose tissue [81]. However, when comparing subcutaneous tissue gene expression for each treatment (control vs. BPA treatment), BPA downregulated several genes involved in immune cell pathways, including those involved in leukocyte activation, inflammatory response, and cytokine production. The physiological basis for immune function gene expression regulation in subcutaneous adipose tissue in animals with BPA prenatal treatment is unknown. These findings may reflect compensatory processes in subcutaneous adipose tissue which overcome metabolic complications, such as the systemic insulin resistance observed in prenatal BPA-treated female sheep. 

Furthermore, Puttabyatappa et al. [82] described in visceral adipose tissue from the mesenteric fat in female sheep prenatally exposed to 0.05, 0.5 and 5 mg/kg/day an increase in IL1β mRNA. Additionally, an increase in MCP1 was evidenced but only at 5 mg/kg/day. This increase in proinflammatory cytokines was accompanied by decreased adiponectin plasma, which is an adipokine with an anti-inflammatory role, and, therefore, its decrease supports the potential proinflammatory state observed in the offspring. However, in this specific work, the comparison of the inflammatory profile of subcutaneous adipose tissue vs. visceral adipose tissue was not evaluated. 

In Refs. [27,76,79,80] additional experiments in mouse models showed the effect of perinatal exposure to BPA (50 µg/kg body weight/day) and its influence on intestinal physiology and the immune system in adult mice (45 postnatal days: PND45) and aged mice (170 postnatal days: PND170). The initial findings were related to an alteration in the Th1/Th17 ratio in the lamina propria and an elevation of Th1/Th17 in the spleen. These modifications were associated with insulin resistance, reduced fecal IgA secretion, and decreased *bifidobacteria* in feces. These features produced by perinatal exposure to BPA were prior to an infiltration of proinflammatory M1 macrophages into the gonadal white adipose tissue, as observed in aged mice. These results reveal that perinatal exposure to this contaminant may have effects later in life [83].

## 7. Conclusions

BPA is a common disruptor and, due to its ubiquitous nature and potential for continued exposure, is detectable in a wide range of body fluids, such as plasma, urine, saliva, breast milk, and amniotic fluid. The dramatic upsurge in the prevalence of obesity has occurred in parallel with an excessive increase in the use of plastic and other products containing endocrine disruptors. Because of their complex interaction with hormone receptors, endocrine disruptor mechanisms of action in the body are difficult to understand. However, in recent years, there has been a growth in experimental research associating BPA exposure to the pathophysiology of obesity, dysregulation of insulin and glucose signaling, and type 2 diabetes mellitus. This relationship is supported by results demonstrating an association between BPA exposure at different doses and increased body weight, alterations in adipokine production, and glucose homeostatic imbalance. It is also clear that this type of disease is related to proinflammatory processes, which aggravate its pathogenesis and prognosis. Interestingly, scientific evidence has begun to show that there is a relationship between BPA exposure and proinflammatory processes in adipose tissue (Figure 3). Nevertheless, the precise role BPA plays as an immunomodulator remains to be elucidated. Moreover, several aspects, such as sex, dose, exposure time, and even the type of response the adipose tissue anatomical compartment might elicit, remain to be clarified. Therefore, further investigations are necessary to gain a more comprehensive view on the association between BPA and inflammation, thus, discerning the possible underlying mechanisms.

## Figures and Tables

**Figure 1 ijms-24-08231-f001:**
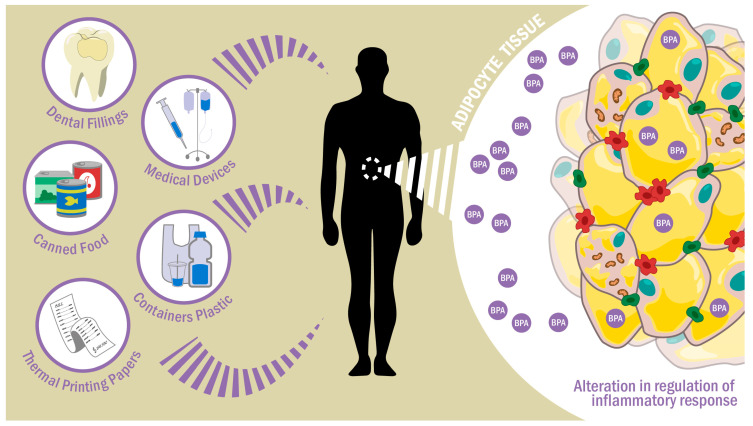
BPA exposure through food, inhalation, dental materials, or dermal exposure. Once it enters the body, it can alter the physiology of adipose tissue by modifying inflammatory pathways.

**Figure 2 ijms-24-08231-f002:**
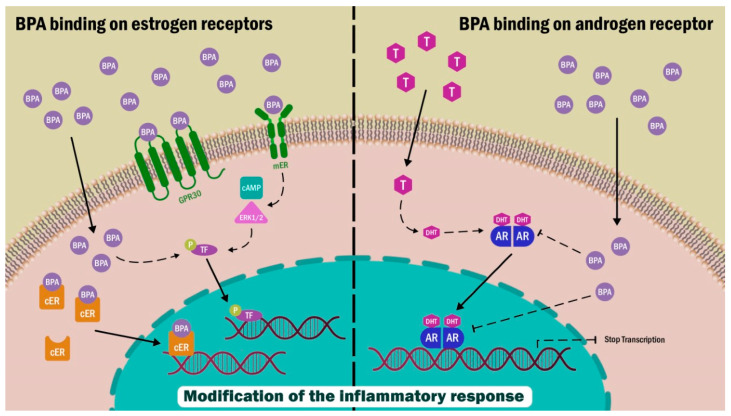
BPA interacts with membrane estrogen receptors (mER), citoplasmatic ERs, GPR30, and androgen receptors (ARs). The effect of this binding can result in an alteration of genomic expression and cell signaling pathways.

**Figure 3 ijms-24-08231-f003:**
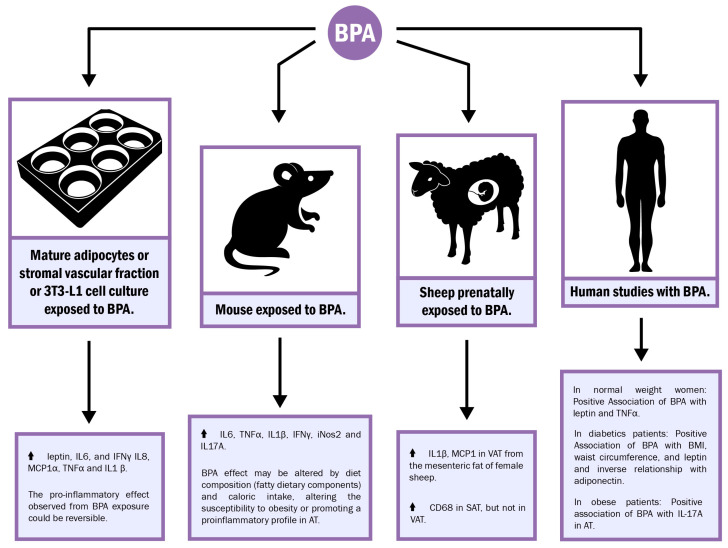
Adipose-tissue-related physiological and cellular consequences of BPA exposure demonstrated in different experimental models. AT: adipose tissue; VAT: visceral adipose tissue; SAT: subcutaneous adipose tissue.

**Table 1 ijms-24-08231-t001:** Urinary BPA levels from human samples.

References (Year)	Type of Participants (n: Number of Participants)	Levels Found	Detection Method	Country
Rebai et al. (2021) [13]	Workers in a plastics industry located in an industrial zone (n: 170).	Average (μg/g creatinine): 3.70.	GC/MS	Algeria
Ayar et al. (2021) [14]	Patients of pediatric intensive care unit: urine samples (n: 115) of children (n: 40).	Mean (μg/g creatinine): 189.2 - First day of hospitalization: 29.5; - The seventh day of hospitalization: 41.1; - After 30 days of hospitalization (or when the patients were discharged): 104.8.	HPLC	Turkey
Aktağ et al. (2021) [15]	Prepubertal children: - Prepubertal children with exogenous obesity (n: 36); - Prepubertal children with exogenous obesity and metabolic syndrome (n: 27); - Control group age- and sex-matched healthy children with no significant underlying medical conditions (n: 34).	Median and (mean ± SD) (μg/g-creatinine): - Prepubertal children with exogenous obesity without metabolic syndrome: 15.0 (25.0 ± 24.2); - Prepubertal children with exogenous obesity and metabolic syndrome: 32.1 (46 ± 39.4); - Control group: 5.0 (6.0 ± 4.6).	LC-MS/MS	Turkey
Chang et al. (2020) [16]	Adults in the Cheyenne River Sioux tribe: American Indians/Alaskan natives (n: 276): - Participants with diabetes (n: 138); - Control group without diabetes matched using age (n: 138).	Geometric mean (μg/L): - Total participants: 1.83; - Participants with diabetes: 1.90; - Control: 1.77.	HPLC-MS/MS	USA
Mohsen et al. (2018) [17]	Children randomly selected from primary and preparatory schools (6–16 years old) (n: 167): - Boys (n: 95); - Girls (n: 72).	Median (ng/mL): - Boys: 0.60; - Girls: 0.67.	HPLC-MS/MS	Egypt
Omran et al. (2018) [18]	Infertile patients presented to the andrology unit (n: 50): - Oligoasthenoteratospermia cases group (n: 16); - Asthenospermia cases group (n: 22); - Asthenoteratospermia cases group (n: 12); - Matched controls with normal semen parameters (n: 50).	Median (µg/g creatinine): - Total infertile cases group: 21.59; - Oligoasthenoteratospermia cases group: 18.16; - Asthenoteratospermia cases group: 19.28; - Asthenoteratospermia cases group: 31.23; - Control group: 19.31.	HPLC	Egypt
Manfo et al. (2019) [19]	Adults between 18 and 59 years of age (n: 81): - Townsmen in urban area: (n: 37); - Farmers using agro-pesticides in rural area: (n: 44).	Arithmetic mean (µg/g creatinine): - All participants: 2.18 ± 1.97; - Townsmen in urban area: 2.16; - Farmers using agro-pesticides in rural area: 2.20.	RIA	Cameroon
Youssef et al. (2018) [20]	Children aged 3–8 years (n: 97): - Asthmatic children (n: 45); - Healthy controls (n: 52).	Median (ng/mL): - Pediatric asthmatic patients: 1.56; - Control group: 0.790.	HPLC-MS/MS	Egypt
Abo El-Atta et al. (2018) [21]	Children/adolescents 2–18 years of age: - Study group: obese children (BMI ≥ 95th percentile) (n: 40); - Control group: normal-weight children (BMI 5th–85th percentile) (n: 40).	Median (min–max) (μg/g creatinine): - Obese children: 121.89 (39.22–586.97); - Control group: 14.92 (<LOD − 34.94).	HPLC	Egypt
Jiménez-Díaz et al. (2016) [22]	Healthy population of women aged 18 years or older (n: 34).	Geometric mean (ng/mL): 0.44. Mean (ng/mL): 1.12.	UHPLC-MS/MS	Tunisia
Keshavarz-Maleki et al. (2021) [23]	- Breast cancer mastectomy patients (n: 41); - Control group: reduction mammoplasty patients with similar BMI to cases group (n: 11).	Mean ± SD (ng/mL): - Breast cancer mastectomy patients: 2.12 ± 1.48; - Control group: 0.91 ± 0.42.	ELISA	Iran
Wu et al. (2021) [24]	Multiethnic cohort (1993–2014): - Postmenopausal women with breast cancer aged 45–75 years: - African American (n: 48); - Latino (n: 77); - Native Hawaiian (n: 155); - Japanese American (n: 478); - White: (n: 274); - Individually matched (n: 1030).	Geometric means (ng/g creatinine): - Whites (n = 547): 1.48; - Japanese Americans (n = 956): 1.07; - Native Hawaiians (n = 309): 1.26; - African Americans (n = 97): 0.77; - Latinos (n = 157) 0.92.	LC/HRAM-MS	USA
Durmaz et al. (2018) [25]	- Newly diagnosed girls with premature thelarche nonobese (aged 4–8 years) (n: 25). - Control group: healthy girls of comparable age with no history of premature thelarche or any other endocrine disorder and no secondary sexual characteristics in their physical exam (n: 25).	Median µg/g (creatinine): - Newly diagnosed girls with premature thelarche, non-obese: 3.21; - Control group: 1.62.	HPLC	Turkey
Radwan et al. (2018) [26]	Males attending infertility clinic for diagnostic purposes with normal semen concentration (n: 315).	Median: 1.87 µg/L, 1.63 µg/g creatinine	GC/MS	Poland
Lee et al. (2019) [27]	Pregnant women who had babies with normal gestation age, neonatal weight, and information on birth outcome: - Neonatal urine (n: 152); - Maternal urine (n: 224).	Median (ng/mL): - Neonatal urine: 4.75; - Maternal urine: 2.86.	HPLC-MS/MS and GC-MS	Korea
Adoamnei et al. (2018) [28]	Healthy, young university students (18–23 years old) (n: 215).	Unadjusted median (ng/mL): 2.8.	UHPLC-MS/MS	Spain
Benson et al. (2021) [29]	Men 18–20 years of age from the Fetal Programming of Semen Quality cohort (n: 556).	Pseudo percentiles (ng/mL): 5th: 0.22, 50th: 1.30, 95th: 9.90.	LC-MS/MS	Denmark
Mínguez-Alarcón et al. (2019) [30]	Women undergoing in vitro fertilization treatment (between 18 and 45 years old) (n: 420).	Geometric means (µg/L): 1.14.	IDMS	USA
Gonzalez et al. (2019) [31]	Workers of a hazardous waste incinerator (n: 29): 11 women and 18 men.	Mean (µg/L): 0.86.	GC/MS	Spain
Hong et al. (2023) [32]	Obese patients and healthy individuals (n: 289): - Obesity cases: participants aged above 16 and below 65 years old, body mass index (BMI) ≥ 27.5 kg/m^2^; - Control group: participants aged above 16 and below 65 years old with body max index < 24.0 kg/m^2^ (n: 152).	Median (µg/g creatinine): - Obesity cases: 4.33; - Control group: 1.37.	LC-MS/MS	China

HPLC: high-pressure liquid chromatography; GC/MS: gas chromatography coupled to mass spectrometry; RIA: radioimmunoassay; LC-MS/MS: high-performance liquid chromatography coupled with tandem mass spectrometry; UHPLC-MS/MS: ultra-high-performance liquid chromatography with tandem mass spectrometry; HPLC-MS/MS: high-performance liquid chromatography (HPLC)–tandem mass spectrometry; ELISA: Enzyme-linked immunosorbent assay; IDMS: isotope dilution high-performance liquid chromatography–tandem mass spectrometry; LC/HRAM-MS: liquid chromatography (LC) with sensitive isotope dilution Orbitrap-based high-resolution accurate-mass mass spectrometry.

**Table 2 ijms-24-08231-t002:** Plasm BPA levels from human samples.

References (Year)	Type of Participants (n: Number of Participants)	Levels Found	Detection Method	Country
Wiraagni et al. (2019) [33]	Healthy volunteers (n: 150): - Males (n: 43); - Females (n: 107).	Observed BPA Levels (ng/mL): Range: 0 to 76.80 Mean: 2.22 ± 9.91 - Males: 0.29; - Females: 2.99; - Less than 33 years of age: 0.847; - 33 years of age and older: 5.852; - Subjects with tap water as source of drinking: 2.882; - Subjects with mineral water as source of drinking: 0.318.	LC-MS/MS	Malaysia
Yamamoto et al. (2016) [34]	Women at 23–35 weeks of gestation and those who delivered between 2002 and 2005: - Maternal blood (n: 59); - Cord blood (n: 285),	Geometric mean (ng/mL): - Maternal blood 0.051; - Cord blood 0.046; Mean (ng/mL): - Maternal blood 0.063; - Cord blood 0.057.	ID-LC/MS/MS	Japan
Pednekar et al. (2018) [35]	Women between 20 and 40 years of age, attending infertility outpatient department, diagnosed with infertility (n: 45): - Polycystic ovary syndrome (n: 31); - Endometriosis (n: 11); - Polycystic ovary syndrome and endometriosis (n: 3); - Married women between 20 and 40 years of age with proven fertility and no evidence of any gynecological disorders, who achieved pregnancy naturally and delivered recently (within one year) (n: 34).	Mean (ng/mL): - Women with infertility: 4.66 ± 3.52; - Polycystic ovary syndrome: 5.80 ± 3.05; - Endometriosis: 4.59 ± 1.22; - Polycystic ovary syndrome and endometriosis (3): 13.17; - Fertile women group: 2.64 ± 3.99.	GC-MS	India
Mas et al. (2018) [36]	Online hemodiafiltration patients using BPA-free (polynephron) or BPA-containing (polysulfone) dialyzers in a crossover design with two arms after a run-in period of at least 6 months with the same membrane: - Patients with BPA-free high-flux polynephron (polynephron) membranes (n: 36); - Patients with high-flux polysulfone (Helixone^®^) dialyzers that contain BPA (n: 36); - Patients on conventional hemodialysis (n: 10); - Healthy controls (n: 10).	Mean (ng/mL): - Patients using BPA-free (polynephron): 8.79 ± 7.97; - Patients with high-flux polysulfone: 23.42 ± 20.38; - Patients on conventional hemodialysis: 98.96 ± 120.75; - Healthy controls: < 2.	High-sensitivity ELISA	Spain
Kolatorova et al. (2018) [37]	Healthy pregnant women of 33 ± 4.1 years (week 37 of pregnancy) (n: 27).	Median with lower and upper quartiles (ng/mL): 0.059 (0.023, 0.084).	LC-MS/MS	Czech Republic
Jin et al. (2017) [38]	Participants of healthy general population, without any evidence of occupational exposure to bisphenols: - Women (n: 9); - Men (n: 10); - Total (n: 19).	Mean (range) (ng/mL): - Women: 0.75 ± 0.12, (0.60–0.88); - Men: 0.60 ± 0.17, range: (0.41–0.85); - Total: 0.67 ± 0.16 (0.41–0.88).	LC-MS/MS	China
Komarowska et al. (2021) [12]	- Children with congenital unilateral cryptorchidism aged 1–4 years (n: 98); - Healthy boys without any disorders of the testes at a comparable age of 1–4 years (n: 19).	Median (ng/mL): - Children with congenital unilateral cryptorchidism: 9.95; - Control group: 5.54.	GC-MS	Poland
Zbucka-Krętowska et al. (2019) [39]	Women undergoing routine amniocentesis between the 15th and 18th weeks of gestation carrying fetuses with a normal karyotype (n: 52).	Mean (ng/mL): 8.69, range: 4.3–55.3.	GC-MS	Poland
Cambien et al. (2019) [40]	Patients suffering from end-stage renal disease and hospitalized (n: 10).	Range (ng/mL): 0.266–86.831.	UHPLC–MS/MS	France
Shen et al. (2016) [41]	- Women with uterine leiomyoma (n: 300); - Control group with no use of hormone drugs during the 3 months prior to the study and free of reproductive-system-related tumors and other estrogen-dependent diseases (breast cancer, endocrine system diseases, etc.) (n: 300).	Media (mean ± SD) (ng/mL): - Women with uterine leiomyoma: 11.19 (16.7 ± 13.9); - Control group: 4.31 (8.62 ± 11.8).	HPLC-MS/MS	China
Ho et al. (2017) [42]	Voluntary human donors (n: 140) age range from 18 to 96 years: - Men (n: 66); - Women (n: 64).	Geometric mean (range) (ng/mL): - Total: 0.53 (range: N.D.–10.43); - Women: 0.59 (N.D.–8.99); - Men: 0.47 (N.D.–10.43).	LC-MS/MS	China
Lin et al. (2017) [43]	- Mother–child pairs with 2-year-old children (n: 208): girls (n: 91), boys (n: 117); - Mother–child pairs with 7-year-old children (n: 148): girls (n: 70), boys (n: 78).	Median (ng/mL): - 2-year-old children: 3.2; girls: 2.9; boys: 3.3; - 7-year-old children: 3.2; girls: 1.2; boys: 4.0.	UPLC-MS-MS	Taiwan

GC/MS: gas chromatography coupled to mass spectrometry; ID-LC/MS/MS: isotopic dilution liquid chromatography–tandem mass spectrometry; ELISA: enzyme-linked immunosorbent assay; UPLC-MS-MS: ultra-performance liquid chromatography–tandem mass spectrometer; LC-MS/MS: high-performance liquid chromatography coupled with tandem mass spectrometry; UHPLC-MS/MS: ultra-high-performance liquid chromatography coupled to a triple quad mass spectrometer; HPLC-MS/MS: high-performance liquid chromatography (HPLC)–tandem mass spectrometry.

## Data Availability

Not applicable.

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
