# Peer review of "New Evidence on BPA’s Role in Adipose Tissue Development of Proinflammatory Processes and Its Relationship with Obesity"

_ijms, 2023, doi:10.3390/ijms24098231_

Round 1
Reviewer 1 Report
Please refer to the attached document.

Author Response
Dear Editor:
We appreciate the rigorous revision of our manuscript.
In the attachment we respond to your suggestions, comments and modifications .

Reviewer 2 Report
I don´t have any especial comment
Author Response
Dear Editor:
We are very grateful and appreciate you taking the time to share your report and feedback with us.
Reviewer 3 Report
In this review the authors address a current and very interesting topic: the influence of bisphenol A on adipose tissue. This is an interesting topic, and the authors put together a lot and recent of information.
Authors evaluated the problem with respect to both the presence of biosphenol A in urine, in plasma and in adipose tissue of different subjects, and investigates its influence on the inflammatory state present in adipose tissue.
The authors report a broad overview of the most recent works in a very specific focus.
The work is well presented and inherent to the topics treated in Int. J. Mol. Sci.
The figures and tables are well structured and clear.
It needs to increase the bibliographic in the general parts.
Add in the conclusion: the importance evaluating the results by sex and gender in future studies.
Here are some general comments.
- Line 137-143; line 144-148, line 167 : there are very long periods without specific references, add the references.
Author Response
Response to Reviewer 3 Comments
Dear Editor:
We appreciate the rigorous revision of our manuscript.
Below we respond to your suggestions, comments and modifications.
- Add in the conclusion: the importance evaluating the results by sex and gender in future studies.
We add the following text to the conclusions
Line 375-378: Nevertheless, the precise mechanism BPA plays as an immunomodulator remains to be elucidated, and how this effect can be modified by factors such as sex or gender.
- Line 137-143; line 144-148, line 167: there are very long periods without specific references, add the references.
We add the following references for each paragraph in accordance with the suggestions given by you
Line 139: Choe SS, Huh JY, Hwang IJ, Kim JI, Kim JB. Adipose Tissue Remodeling: Its Role in Energy Metabolism and Metabolic Disorders. Front Endocrinol (Lausanne). 2016 Apr 13;7:30. doi: 10.3389/fendo.2016.00030. PMID: 27148161; PMCID: PMC4829583.
Line 141: Kawai T, Autieri MV, Scalia R. Adipose tissue inflammation and metabolic dysfunction in obesity. Am J Physiol Cell Physiol. 2021 Mar 1;320(3):C375-C391. doi: 10.1152/ajpcell.00379.2020. Epub 2020 Dec 23. PMID: 33356944; PMCID: PMC8294624.
Line 167: Araiza VHDR, Mendoza MS, Castro KEN, Cruz SM, Rueda KC, de Leon CTG, Morales Montor J. Bisphenol A, an endocrine-disruptor compund, that modulates the immune response to infections. Front Biosci (Landmark Ed). 2021 Jan 1;26(2):346-362. doi: 10.2741/4897. PMID: 33049673.
Round 2
Reviewer 1 Report
The authors have addressed the comments and suggestions as far as this reviewer is concerned.